# Estimation of Tinnitus-Related Socioeconomic Costs in Germany

**DOI:** 10.3390/ijerph191610455

**Published:** 2022-08-22

**Authors:** Konstantin Tziridis, Jana Friedrich, Petra Brüeggemann, Birgit Mazurek, Holger Schulze

**Affiliations:** 1Experimental Otolaryngology, University of Erlangen-Nuremberg, Waldstrasse 1, 91054 Erlangen, Germany; 2Tinnitus Center, Charité University Medicine Berlin, Luisenstrasse 13, 10117 Berlin, Germany

**Keywords:** tinnitus, hearing loss, socioeconomic costs, tinnitus questionnaire, health utility index

## Abstract

Despite the high prevalence of tinnitus in Germany of nearly 12% of the general population, there have been no systematic studies on the socioeconomic costs for German society caused by tinnitus so far. Here we analyzed data from 258 chronic tinnitus patients—namely tinnitus severity and health utility index (HUI)—and correlated them with their tinnitus-related public health care costs, private expenses, and economic loss due to their tinnitus percept as assessed by questionnaires. We found correlations of the HUI with health care costs and calculated the mean socioeconomic costs per tinnitus patient in Germany. According to our most conservative estimate, these sum up to EUR 4798.91 per year. Of that EUR 2206.95 account for the public health care, EUR 290.45 are carried by the patient privately and the remaining EUR 2301.51 account for economical loss due to sick leave. With a prevalence of 5.5% with at least bothersome tinnitus, this sums up to 21.9 billion Euro per year and with 25.82 sick leave days; tinnitus patients miss work more than double the time of the average German employee (10.9 days). The findings fit within the cost ranges of studies from other European countries and the USA and show that the socioeconomic burden of this disease-like symptom is a global problem. In comparison with the costs of other major chronic diseases in Germany—such as chronic obstructive pulmonary diseases (ca. 16 billion Euro) or diabetes mellitus (ca. 42 billion Euro)—the relevance of the ‘symptom’ tinnitus for the German social economy becomes even more obvious.

## 1. Introduction

In Germany, up to 25% of all patients showing up in ear, nose, and throat (ENT) practices suffer from tinnitus [1]. The prevalence in the general population is assumed to be at nearly 12%, with 5.5% reporting at least bothersome tinnitus [2]; an increased prevalence for tinnitus in the elderly can be found [3]. The patients often suffer more strongly from the symptom tinnitus than from the disease it originates from [4]. Vice versa, population studies also revealed, e.g., that tinnitus patients suffer stronger from hearing loss [5], hyperacusis [6,7], or reduced speech intelligibility [8,9] and this might be even more pronounced in the elderly [10]. Additionally, the phantom sound can lead to insomnia, psychological disorders, or—for the most severe cases—even suicide attempts [11,12,13], all leading to the need of additional medical attendance.

Despite the huge numbers of patients affected by this symptom and major worldwide efforts to investigate the cause and treatment of tinnitus [2,14,15,16,17,18,19,20,21,22,23,24,25,26,27,28], only very few studies on the socioeconomic costs of tinnitus itself have been carried out [29,30,31]. One European example is the Dutch study of Meas and colleagues [32], who found the yearly socioeconomic costs of 2012 for one tinnitus patient to be around EUR 5250, summing up to annual socioeconomic costs of around EUR 6.8 billion in The Netherlands. We tried to replicate the questionnaires used in that study as much as possible to be able to make a comparison of the Dutch and German costs, as the Dutch healthcare system is often referred to as a model system for Germany [33] and the salary level structures are comparable. The described costs in the Meas study are significantly higher than those for other major diseases with comparable prevalence in that country, e.g., for borderline personality disorders (EUR 2.2 billion) [34], social phobia (EUR 1.7 billion) [35], or low-back pain (EUR 3.5 billion) [36].

With this study, we aimed to investigate the costs for tinnitus for the German healthcare system, the private costs taken by each patient, and the economic loss due to the symptom. To this end, we analyzed questionnaires specifically designed to answer these questions in tinnitus patients. The questionnaires were obtained in the years 2016 and 2017 in 258 patients of the Berlin Charité Tinnitus center as part of one of the largest hospitals in Europe and therefore capable of recruiting a large number of patients.

## 2. Material and Methods

### 2.1. Ethical Statement

The study was approved by the ethics committee of the Charité university hospital Berlin (EA4/137/20). A group of 258 chronic tinnitus patients (140 female, 54%) with a mean age (±standard deviation) of 52.3 ± 10.7 years were examined at the tinnitus center of the Charité hospital in the years 2016 and 2017. Exclusion criteria were insufficient knowledge of German language, acute psychoses, or motoric impairments.

### 2.2. Data Acquisition

On their first visit to the tinnitus center, patients received the tinnitus cost questionnaire (TCQ, cf. Appendix A, with English translations) with 70 questions in German. The TCQ consisted of two parts: The first, larger part included 52 questions regarding previous medical examinations of the last three months that were in correlation with the tinnitus percept to assess all health care services and treatments used by the patients. Included in this part were also questions regarding self-treatment, alternative medication, and private expenses, e.g., for public transportation or taxi rides in this context. The second part, consisting of 18 questions, aimed to evaluate the income situation of the patients as well as how many sick leave days in the last three months had to be taken because of the tinnitus percept.

Additionally, the patients were handed the standardized health utility questionnaire (HUI) for assessing the general physical health and quality of life [37]. The HUI has several subscores, which we investigated additionally to the overall score of the questionnaire (cf. Statistics). Third, the standardized tinnitus questionnaire (TQ) of Goebel and Hiller [38] assessed the tinnitus severity. Its results were used to group the patients into the four tinnitus severity groups for further analysis (cf. Statistics). Finally, hearing loss (HL) data of both ears were collected within the framework of general hearing diagnostics. Included frequencies were 0.25 kHz, 0.5 kHz, 1 kHz, 1.5 kHz, 2 kHz, 3 kHz, 4 kHz, 6 kHz, and 8 kHz.

### 2.3. Calculation of Socioeconomic Costs

Primary costs for the public health care system per quarter were doctors’ visits and treatment costs. Each visit with a general practitioner, specialist, or other health service provider (e.g., medical officier, social worker, physiotherapist, audiologist and many more)—analogously to the earlier Dutch study [32]—was calculated according to the health insurance status (public or private health insurance, cf. Appendix A) of the patient. Calculations were based on the most recent cost tables for general practitioners, specialist, dentists, psycho- and physiotherapists, etc. Costs for prescribed drugs specifically for tinnitus-related treatments were calculated accordingly. For patients whose status of their health insurance was missing or unclear, we assumed public health insurance. After the calculation of the individual costs, the mean health care system costs per quarter for the patients—depending on their tinnitus severity—were calculated and extrapolated for one whole year to obtain the annual costs.

Similar to the described approach, the private expenses for each patient and the complete sample were calculated. This included public or private transport to and from doctors’ offices, as well as costs for non-standard health care providers (e.g., hypnotists, acupuncturist), nonprescription drugs, sport, meditation sessions, and other expenses. For the transport cost calculations, we used the information from the TCQ (car, bicycle, public transport, by foot, other) and assumed the mean distance to the nearest general practitioner (2977 m) and specialist (8485 m) in Germany [39]. The distance was multiplied with the tax kilometer rate of EUR 0.30 for car transport. For public transport, the costs were set to EUR 2.90 per direction according to the single ticket tariff in Berlin, Germany. Transport by bicycle, by foot, or via other methods were not included into these costs. Again, after the calculation of the individual costs, the mean private costs per quarter for the patients dependent on their tinnitus severity were calculated and extrapolated for one whole year.

For the estimation of the economic costs of tinnitus, the number of sick days and the salary of the patients in one quarter were evaluated. The number of work days per month was set to 21.75 according to German wage tax guidelines [40]. By dividing the individual monthly salary by 21.75, the salary for one day was calculated and the economic loss through sick days per quarter could be estimated. Again, a mean value of sick days and salary loss per year was calculated. To compare the economic costs of tinnitus with the costs for other major chronic diseases, the loss in gross value was calculated. For this purpose, the number of sick days per quarter was multiplied by EUR 203 according to the “Bundesanstalt für Arbeitsschutz und Arbeitsmedizin” [41] and estimated for one year. For the estimation of the complete socioeconomic costs of tinnitus, all three described annual values were added up.

### 2.4. Statistics

The TCQ reliability was tested with Cronbach’s Alpha and returned a value of α = 0.65, which is acceptable. In addition to the described calculations, we performed non-parametrical participant statistics for the characterization of the patient population. We here focused on the patients’ tinnitus severity, age, gender, and level of education. This was performed in accordance with the study from Maes et al. [32] where the level of education was separated into three categories ‘low’, ‘medium’, and ‘high’.

We further analyzed the dependence of the binaural HL on ‘frequency’ and ‘tinnitus severity’ by a two-factorial ANOVA, as this variable is normally distributed. We non-parametrically characterized the mutual dependencies of the HL, tinnitus severity, and HUI score by multiple linear regressions. In addition, we correlated the tinnitus severity and the HUI score (and HUI sub-scores) with the socioeconomic costs of tinnitus, as well as with the three different categories of tinnitus costs by multiple linear regressions. As an additional reliability check, these data were also analyzed with a general linear model approach. Finally, we compared the different variables in compensated (severity index 1 and 2) and uncompensated tinnitus (severity index 3 and 4) patients by Mann–Whitney U-tests.

## 3. Results

### 3.1. Patient Statistics

Of the 258 patients participating in the study (cf. Table 1), most showed low tinnitus severity indices (severity index 1: 90 patients (34.7%), severity index 2: 82 patients (31.7%)), while severity index 3 was reported by 56 patients (21.6%), and severity index 4 by 30 patients (11.6%). Among participants, 172 patients had compensated tinnitus (66.7%) and 86 uncompensated tinnitus (33.3%). The mean age of all patients was 52.3 ± 10.7 years with no significant age differences between the four tinnitus severity groups (Kruskal–Wallis ANOVA, H (3, N = 257) = 2.15, *p* = 0.54: severity index 1: 51.0 ± 11.7 a; severity index 2: 53.5 ± 10.9 a; severity index 3: 52.2 ± 10.4 a; severity index 4: 53.7 ± 7.6 a). Furthermore, in the gender distribution (54% female), no significant difference in the chi-square test for multiple groups (Χ^2^ (3, N = 257) = 1.73, *p* = 0.63) could be found for the four different tinnitus severity patient groups. Among the patients, 3.6% had a ‘low’ educational level, 28.7% had a ‘medium’ educational level, and 67.7% had a ‘high’ level of education.

### 3.2. Audiological and HUI Dependencies

For the investigation of the dependencies of the different variables (tinnitus severity, binaural HL and HUI score), we first focused on the ‘audiology results’, i.e., the binaural HL and the tinnitus severity index. We analyzed the binaural HL of the patients’ audiograms in relation to their individually determined tinnitus severity index by a three-factorial ANOVA (factors ‘frequency’ and ‘tinnitus severity’ index). We found the ‘classic’ effect of high frequency binaural HL averaged across all severities (frequency: F(8, 4545) = 225.20, *p* < 0.001) with low frequencies showing hardly any HL (e.g., 0.25 kHz: 12.8 ± 10.9 dB) while high frequencies showed a moderate HL (e.g., 8 kHz: 41.4 ± 23.0 dB) averaged across all patients. The tinnitus severity also had a strong effect on the binaural HL (Figure 1A, inset; F(3, 4545) = 100.1, *p* < 0.001) with each increase in severity leading to a significant increase in HL (Tukey post-hoc tests, always *p* < 0.001). Both factors did not show any interaction (Figure 1A; F(24, 4545) = 0.57, *p* = 0.95), indicating a parallel shift in the audiograms dependent on the tinnitus severity index. This dependency of the HL on the tinnitus severity was also indicated by the significant positive linear correlation of mean binaural HL (averaged over all frequencies) and the tinnitus severity index shown in Figure 1B (multiple linear regression analysis: r^2^ = 0.11, *p* < 0.001).

To correlate the results of the HUI questionnaires with the ‘audiological results’ described above, we performed multiple linear regression analyses of the HUI score with the tinnitus severity index (Figure 1C) and the mean binaural HL (Figure 1D). We found in both analyses bore significant positive linear correlations (both *p* < 0.001) with an especially high regression coefficient of r^2^ = 0.89 in the HUI/tinnitus severity correlation, while the coefficient of the HUI/HL correlation only reached r^2^ = 0.10. In other words, the general physical health and quality of life of the patients was especially more strongly impaired with higher tinnitus severity.

### 3.3. Socioeconomic Cost of Tinnitus

For estimating the costs of tinnitus for the health care system we evaluated the mean costs for the single visits of the different medical services used (Table 2; Appendix A). We analyzed how many times each service was used by each patient in one quarter and extrapolated this to one year. In most cases, medical services produced different mean costs for public and private insurance and were calculated appropriately following the German “Einheitlicher Bewertungsmaßstab” (EBM) [42] and the “Gebührenordnung für Ärzte” (GOÄ) [43]. The mean cost values for the different medical services were calculated based on different scenarios regarding the insurance status, the age of the patient, the equipment used, and if it was the first visit or a follow-up visit within the quarter.

For the visit at a general practitioner, we therefore took six possible scenarios into account, where the insurance flat rate regarding the age of the adult patient was already averaged, based on the EBM (below 55 a: EUR 13.20; below 76 a: EUR 16.99; from 76 a: EUR 22.84; used average: EUR 17.68). Scenario 1: public health insurance patient at first visit (EUR 17.68) with audiometer examination (EUR 9.52; EU directive 93/42/EWG) summing up to EUR 27.20. Scenario 2: public health insurance patient at first visit without audiometer examination accounting to EUR 17.68. Scenario 3: public health insurance patient at follow-up visit (no insurance flat rate accountancy) with audiometer examination summing up to EUR 9.52. Scenario 4: public health insurance patient at follow-up visit without audiometer examination, leading to no compensation. From these four scenarios, the mean public health insurance costs (EUR 13.60) were calculated. Scenario 5: private health insurance patient (based on GOÄ) at any visit without specific examination (EUR 20.11) and general examination (EUR 13.41) summing up to EUR 33.52. Scenario 6: private health insurance patient at any visit (EUR 16.58) but with specific ear examination (EUR 10.72) and otoscopy (EUR 13.41) summing up to EUR 40.71. From these two scenarios, the mean values for private insurance costs (EUR 37.12) were calculated.

The cost calculation for the house calls for general practitioners was also based on six scenarios. Scenario 1: public health insurance patient at first visit (EUR 17.68) with audiometer examination (EUR 9.52), house call flat rate (EUR 22.94), and travel expenses. These travel expenses differed dependent on the public health insurance company and on the distance traveled (zones). The mean travel expenses for the zone 1 (beeline 2 km or less) were calculated with EUR 4.30, for zone 2 (beeline 2 to 5 km) with EUR 8.73 and for zone 3 (beeline maximal 10 km) with EUR 12.68, averaging to a value of EUR 8.57. In summation, the costs for this scenario were EUR 58.71. Scenario 2: public health insurance patient at first visit (EUR 17.68) without audiometer examination, house call flat rate (EUR 22.94), and travel expenses (EUR 8.57), summing up to EUR 49.19. Scenario 3: public health insurance patient at follow-up visit with audiometer examination (EUR 9.52), house call flat rate (EUR 22.94), and travel expenses (EUR 8.57), summing up to EUR 41.03. Scenario 4: public health insurance patient at follow-up visit without audiometer examination, house call flat rate (EUR 22.94), and travel expenses (EUR 8.57), summing up to EUR 31.51. From these four scenarios, the mean public health insurance costs (EUR 45.11) were calculated. Scenario 5: private health insurance patient at any visit without specific examination (EUR 42.90) and travel expenses (zone mean of EUR 8.95) summing up to EUR 51.85. Scenario 6: private health insurance patient at any visit (EUR 16.58) but with specific ear examination and otoscopy (EUR 42.90) and travel expenses (EUR 8.95), summing up to EUR 68.43. From these two scenarios, the mean values for private insurance costs (EUR 60.14) were calculated.

The calculation for the emergency service costs was based on four scenarios. Scenario 1: public health insurance patient visit outside of classic workday times were accounted with EUR 21.10 [42]. Scenario 2: private health insurance patient visit [43] outside the classical worktimes of workdays were accounted with EUR 44.01. Scenario 3: private health insurance patient visits during daytime at the weekend summing up to EUR 46.34. Scenario 4: private health insurance patient visits during nighttime at the weekend summing up to EUR 56.83. From these three last scenarios, the mean values for private insurance costs (EUR 49.06) were calculated.

ENT specialist visit-related costs were estimated from three scenarios. Scenario 1: public health insurance patient first time visit basic flat rate was dependent on the age of the patient. Below 60 years the value was accounted with EUR 21.43, with 60 years and above the value was EUR 22.19, averaging to EUR 21.81. The examination for the ENT diagnostic summed up to EUR 45.57, which in turn summed up with the basic flat rate to EUR 67.38. Scenario 2: public health insurance patient follow up visit with examinations accounted to EUR 45.57. Both scenarios resulted in the average public health care cost of EUR 56.48. Scenario 3: private health care insurance patient at any visit with all suggested examinations following GOÄ accounted for EUR 74.62.

Alternatively to a visit at an ENT specialist, tinnitus patients also visited ENT hospitals for ambulant examination and treatment. The cost estimation was based on two scenarios. Scenario 1: public health insurance patient at any visit flat rate cost accounted to EUR 97.50. Scenario 2: private health care insurance patient at any visit followed the GOÄ [43] and summed up to EUR 138.05.

For further clarification of possible neurological disorders, visits at the neurologist might be useful for the patients. Neurologist visit-related expenses for the health care system were estimated from three scenarios. Scenario 1: public health insurance patient first time visit basic flat rate was dependent on the age of the patient as described above, the average costs were EUR 24.79. A wide range of examinations could be performed following the EBM [42] that may sum up to EUR 150.11, but the examinations most probably would be split into at least two separate examination days, which makes the estimation of this point somewhat difficult. We assumed two days for a complete examination and therefore calculated one visit with (EUR 24.79 + EUR 150.11)/2 = EUR 87.45. Scenario 2: public health insurance patient follow-up visit in the quarter. Here, only the costs for the complete examination over two separate days accounted with EUR 150.11/2 = EUR 75.06. Both scenarios resulted in the average public health care cost of EUR 81.25. Scenario 3: private health care insurance patient at any visit with all suggested examinations following GOÄ accounted over two examination days for EUR 424.11. Per visit, this accounted for EUR 212.06 for the private health insurance cases.

After the visit with an ENT specialist or hospital with or without an examination by a neurologist, many tinnitus patients need hearing aids. Usually, three visits at the audiologist were needed for the fitting of a hearing aid. Each audiologist has fixed contracts with the different health insurance providers setting the ‘standard costs’ for predefined hearing aids that are covered completely by the health insurance. The difference for more expensive devices had to be covered by the patients privately [44]. We here estimated the costs from two scenarios. Scenario 1: public health insurance patient with ‘standard device’. The mean covered costs over the known providers (e.g., [45,46]) were EUR 766. With three visits the average visit costs were EUR 255. Scenario 2: private health care insurance patient with average covering [47] could expect up to EUR 1500. With also three visits, the average covered costs were EUR 500 per visit.

For tinnitus co-morbidities, visits at the psychiatrist could be useful for some tinnitus patients. Mean expenses for psychiatrist visits by tinnitus patients were based again on three scenarios. Scenario 1: public health insurance patient first time visit basic flat rate was dependent on the age of the patient as described above, the average costs were EUR 21.43. The basic flat rate included 10 min of psychological counselling. We assumed that this would not be enough time for the complete counselling sessions and therefore added the counselling fee of EUR 14.72 for sessions exceeding 20 min to the basic costs, summing up to EUR 36.15. Scenario 2: public health insurance patient follow-up visit in the quarter. Here, only the costs for the counselling session of EUR 14.72 could be taken into account. Both scenarios resulted in the average public health care cost of EUR 25.44. Scenario 3: private health care insurance patient at any counselling visit would account to EUR 67.04.

Only a few tinnitus patients visit a psychiatrist; and if so, they are diagnosed with any affective or stress-related disorder only in 50% of the cases [48]. Visits with psychotherapists can be useful for them. Again, the mean cost estimations were derived from the three scenarios already mentioned above. We only took treatment options without the need for specific application at the health care provider into account. Scenario 1: public health insurance patient first time visit basic flat rate was dependent on the age of the patient as described above, the average costs were EUR 12.23. Six sessions per quarter at EUR 50.00 were taken over by the insurance plus four additional probatory sessions in the case of a suspicion of a disorder summing up to EUR 268.83. After this, we estimated 24 sessions at EUR 99.78/session for a bridge therapy, as this might be the most common approach for tinnitus patients. All these costs summing up to EUR 3029.15 for all 35 sessions, for a single session we therefore calculated EUR 86.55. Scenario 2: public health insurance patient follow-up visit in the quarter with all 35 sessions summing up to EUR 3016.92 or EUR 86.20/session. Both scenarios resulted in the average public health care cost of EUR 86.38 per session. Scenario 3: private health care insurance patients were treated usually over 29 therapy sessions summing up to EUR 3123.06 or EUR 107.69 per session.

Next to psychiatrists and psychotherapists, also visits with an occupational therapist might be useful for tinnitus patients to treat co-morbidities such as anxiety disorders or depression, on which this study focused. The costs for a visit with an occupational therapist were based on the therapeutic products catalogue [49] and were based on two scenarios. Scenario 1: public health insurance patient at any visit with psychological-functional treatment accounted for EUR 65.72 per session. Scenario 2: private health insurance patient at any visit with psychological-functional treatment accounted for EUR 118.30 per session.

In the literature, there are hints that tinnitus patients might have an elevated prevalence for a craniomandibular dysfunction (CMD) [50]. Therefore, visits at the dentist might also be useful for these patients. The costs for these visits were taken from the “Bewertungsmaßstabs für zahnärztliche Leistungen” [51] and estimated from four scenarios. Scenario 1: public health insurance patients’ anamnesis without positive CMD diagnosis accounted for EUR 19.49. Scenario 2: public health insurance patients’ anamnesis with positive CMD diagnosis accounted first with the initial EUR 19.49, then with EUR 21.65 for the preparation of a treatment and cost plan for the therapy with a bite rail. The mean lab costs of EUR 213.89 and the further treatment costs of EUR 142.91 plus the EUR 19.49 summed up to EUR 376.29 for the complete treatment. The usual number of visits was found to be three, which results in a single session cost of EUR 125.43. Both scenarios resulted in average dentist costs of EUR 72.46. Scenario 3: private health insurance patient‘s anamnesis without positive CMD diagnosis accounted for EUR 23.66. Scenario 4: private health insurance patient’s anamnesis with positive CMD diagnosis accounted with mean treatment and lab costs for EUR 542.59 over three sessions, resulting in a single session cost of EUR 180.86. Both private health care insurance scenarios resulted in the average dentist costs of EUR 102.26.

In the case of a somatic tinnitus, physiotherapy might be a suitable therapeutic approach [52,53,54]. The costs for a visit at a physiotherapist were based on the therapeutic products catalogue [55] and only included manual and thermo-therapy. Here, only two scenarios were taken into account. Scenario 1: public health insurance patient at any visit with manual therapy (EUR 25.35) and thermo-therapy (EUR 5.29) summing up to EUR 30.64. Scenario 2: private health care insurance patient at any visit. The estimation is quite difficult here, as no legal boundaries are defined. We use the evaluation of Buchner from 2019 [56] as the basis of our estimation. Therefore, manual therapy accounted for EUR 42.03 and the thermo-therapy for EUR 9.52, summing up to EUR 51.55 per session.

Social worker costs could have emerged when tinnitus patients needed support—e.g., after hospital treatments or at counselling centers—and are therefore based on a large variety of funding sources. As these costs could not be specifically calculated, we used the average 38 h week monthly salary based on the WSI-wage level database [57] of EUR 2827 and calculated the working hour costs with EUR 17.17. We estimated the average counselling duration to be half an hour resulting in costs of EUR 8.59 per session, independent of the insurance status of the patient.

Similar to the described estimation problem above, the costs for a visit at the medical officer could also not be calculated directly. We used the average yearly salary of a medical officer of EUR 84,200 as basis and calculated the average working hour costs with EUR 40.48. We estimated the average counselling duration to be 10 min. The average costs therefore accounted for EUR 6.75 per visit, independent of the insurance status of the patient.

Finally, some patients noted visits at other specialists or alternative medicine approaches, partially or completely covered by some health care providers. These visits were evaluated following the EBM or the GOÄ. For an overview of these costs, refer to Table 3.

For the estimation of the complete costs for all doctors’ office visits, the sum of the visits for a given visit class for patients with public or private health insurance, dependent on the severity of their tinnitus percept (indices 3 and 4 were combined), were multiplied with the respective estimated costs per visit per quarter. All costs were added up, multiplied by four, and divided by the number of patients to obtain the mean costs per patient per year (cf. Table 3). The costs accounted for EUR 2178.67 per year.

Costs for prescribed drugs specifically for the patients’ tinnitus were evaluated from the information given in the TCQ. An overview of the prescribed active ingredient, pack size, and price is given in Table 4. Of the 258 investigated patients, 57 (22.1%) specified that they used prescribed drugs in the last quarter, but only 33 patients (12.8%) were able to identify them from the list provided in the TCQ or by writing the active ingredient or drug name explicitly. To estimate the mean drug costs per capita, the cost of one pack of the drug was multiplied by the number of patients using it. All costs were then summed up (EUR 1055.84) and divided by 33, resulting in EUR 32 per capita and quarter. The remaining 24 drug using patients were added with this mean value to obtain the average total costs of EUR 1823.84 for all 57 patients and therefore for every one of the 258 patients an average of EUR 7.07 drug costs per quarter, or EUR 28.28 per year. Together with the above estimated costs for the visits at the doctors’ offices, the complete costs for the German health care system added up to EUR 2206.95 per patient and year spread over 38.92 visits.

For the estimation of the costs to be borne privately, first we accounted for over-the-counter drugs at EUR 1525.95 per quarter used by 20% of all patients. This resulted in average per capita costs of EUR 23.66 per year. Second, we summed up the expenses for sports or meditation, which together used 32% of all patients to counteract their tinnitus percept. The costs for this were EUR 6115 per quarter, or EUR 94.80 averaged per capita per year. Third, the travel costs to and from the doctors’ office visits were calculated as described in the Methods section. This estimation resulted in total travel costs of EUR 66.14 per capita annually. Finally, other expenses in the last quarter were noted from 12% of all patients, ranging from EUR 6 to EUR 2700 with a total sum of EUR 6852.90, resulting in yearly per capita expenses of EUR 105.84. In sum, the private costs of tinnitus were EUR 290.45 per capita annually.

The estimation of the economic costs of tinnitus were mainly accounted for by the loss of labor and productivity through sick days. For that purpose, we analyzed the number of sick days in the last quarter and the total net income of each patient. The daily salary was then calculated following the German income tax directive of a 5-day workweek with an average 21.75 workdays per month. From the total quarterly individual economic costs, we calculated the average of EUR 2301.51 per capita annually, with a mean number of sick days of 25.82 days per year. For comparison, the mean number of sick days in Germany in 2019 was 10.9 days [58].

Another key figure for the economic costs of the incapacity to work is the lost gross value added, which represents the loss of work productivity by incapacity to work. In 2017, the value for the loss of gross value added per day of incapacity to work was EUR 203 [41]. Multiplying this value with the number of sick days per year, the economic costs reached a value of EUR 5240.67 per capita annually. This value can be seen as the maximum of the estimation for the economic loss, while the individually calculated value of EUR 2301.51 might represent the minimum estimated loss.

By adding up the three main categories of the socioeconomic costs of tinnitus, we can estimate a minimum and maximum mean individual range. The “fixed” estimations for the health care system (EUR 2206.95) and the private costs (EUR 290.45) can be summed up with either the minimal estimated economic loss value of EUR 2301.51 or the maximum value of EUR 5240.67 resulting in costs of either EUR 4798.91 or EUR 7738.07 per capita annually. The prevalence for bothersome tinnitus in Germany is 5.5%, with an estimated 83,100,000 inhabitants (status 2021) the number of patients amounted to 4,570,500 people. Therefore, the yearly socioeconomic costs for tinnitus accounted for between EUR 21,933,418,155 and EUR 35,366,848,935 in Germany.

### 3.4. Dependencies of the Socioeconomic Costs on Other Variables

As indicated in Table 3, we were not only interested in the ‘raw’ costs, but also in the dependencies of the different costs on dependent variables such as the tinnitus severity (obtained with the tinnitus severity index) or the HUI score/sub-scores of the individual patients. For that purpose, we performed several multiple linear regression analyses that are summed up in Table 5. Generally, we found the majority of the significant regressions to be with the individual healthcare costs. The individual overall socioeconomic costs were neither linearly correlated with the tinnitus severity index (Figure 2A) nor with the HUI score (Figure 2B). The individual healthcare costs, on the other hand, showed a tendency for a linear correlation with the tinnitus severity index (Table 5), as well as a significant linear correlation with the HUI score (Figure 2C) as well as with three of the six HUI sub-scores (Figure 2D). Nevertheless, all regressions were weak as indicated by the low r^2^ values not exceeding values of 0.04. The two cases where at least a tendency for a linear correlation of costs with HUI sub-scores were identified were: first, the individual overall costs with the sleep sub-score; and second, the individual private costs with the somatic sub-score (Table 5). For the individual economic costs, no significant linear correlations were found at all. To check for reliability of these analyses and to find possible weak effects, we repeated the HUI data analyses with a general linear model approach. The results of these analyses are summarized in Table 6; generally, we did not find significantly different results from the analyses described above, with the exception of losing the tendency for a correlation of severity index and healthcare costs.

To identify possible nonlinear dependencies of the costs on the compensation status of the tinnitus (compensated (severity index 1 and 2) and uncompensated (severity index 3 and 4)), we calculated nonparametric Mann–Whitney U-tests for the overall socioeconomic costs and the three cost sub-classes. We found significant dependencies of the overall costs as well as the healthcare and private costs on the tinnitus compensation status, with uncompensated tinnitus showing higher costs for the patients than for patients with compensated tinnitus. No significant differences were found for the economic costs between the two different tinnitus patient groups (Table 7). As the economic costs are strongly dependent on the individual profession, we here focused our analysis on the number of sick days both patient groups stated in the questionnaire. We found a significantly higher number of sick days in the uncompensated compared to the compensated tinnitus patients (Mann–Whitney U test: compensated 0 [0, 12], uncompensated 25 [0, 45], *p* < 0.001).

## 4. Discussion

With this study, we were for the first time able to estimate the yearly socioeconomic costs of tinnitus in Germany. We could show a direct correlation of increased healthcare costs with an increase in impaired quality of life (HUI), which in turn strongly correlates with the tinnitus severity. In addition, the number of sick days were significantly higher in severely affected tinnitus patients compared to patients subjectively perceiving only a mild tinnitus. The annual socioeconomic costs were estimated as being between EUR 21.9 and EUR 35.3 billon for the 5.5% of the German population that are at least mildly affected tinnitus patients.

Tinnitus is usually seen as a symptom of stress or hearing loss [26,59] and has a prevalence of 11.9% in Germany [2]. Only roughly half of those perceiving such a tinnitus percept report it as at least bothersome; the prevalence of such a percept has been found to be 5.5% of the total population. Of those, the majority (66.6%)—at least in our sample—seem to be able to compensate the percept to a degree that they are able to “live with it”. The remaining third of the patients suffer from uncompensated tinnitus and have a much higher intrinsic pressure to counteract their percept. Those patients have a significant higher number of sick days compared to the compensated tinnitus patients and produce higher costs in the healthcare sector (cf. Figure 2C,D). This may be due to an increased number of comorbidities, especially in psychiatric diseases [48], leading to these higher costs.

The estimation of the different costs for our study could be mispriced due to several reasons. First, the patient group of 258 individuals might not be representative for the tinnitus patient population. On the other hand, the mean age of 52.3 ± 10.7 years and a nearly equal distribution of male and female participants (54% female) fitted well within the range of other studies (e.g., [60,61,62]). Second, we could have underestimated the individual costs for specific healthcare categories for the chronic tinnitus patients, as we only asked about the last three months and calculated the yearly costs from that mean value for the complete patient group. Third, this calculation of the mean value over all healthcare categories has to be seen as only a rough estimator for the true costs. We do not know the exact weight each category has for the whole population of tinnitus patients. Fourth, the healthcare and private costs might not be complete, as some patients might have not included all visits or all purchases they made for many possible reasons. We were not able to correct for such errors. Fifth, and last, the economic costs might be underestimated, as we did not receive answers form all patients in the relevant questionnaire fields and several patients (42/258, 16.3%) did not declare any income at all. Our conservative yearly socioeconomic cost estimation of EUR 21.9 billion (for 2017) is roughly triple the value of the inflation-corrected (mean European inflation 2012 to 2017: 1%) Dutch study [32], which was used as a reference for our approach. Our data acquisition and cost estimations were as near as possible to that study. The Dutch study would today have a calculated volume of EUR 7.2 billion for the 17.4 million inhabitants of the Netherlands. With its 83.1 million inhabitants, the estimated maximum costs of EUR 35.3 billion in Germany seems to be more in line with the inhabitants number ratio of both countries (inhabitants GER/NL = 4.8; maximum costs GER/NL = 4.9), indicating that—most probably—that higher value might reflect the true socioeconomic costs of tinnitus.

A recent review of socioeconomic costs of tinnitus in the USA, UK, and The Netherlands (i.e., our reference study) from 2021 [29] reported annual costs per patient between EUR 1544 and EUR 3429 for the healthcare system, between EUR 69 and EUR 115 for private expenses and between EUR 2565 and EUR 3702 in indirect costs including productivity loss. In comparison with our results of annual costs of EUR 2207 for the German healthcare system—EUR 290 in private costs and an estimated annual economic loss of between EUR 2302 and EUR 5241—the above mentioned costs seem to be within the same range but would have to be corrected for differences in the healthcare and salary levels of each country. This is beyond the focus of this study.

Independent of the exact value of the socioeconomic costs, even the conservative estimate of EUR 21.9 billion per year shows that the costs of tinnitus with a prevalence of 5.5% are comparable with those of other major diseases in Germany. Some prominent examples for such diseases would be—on the lower end—chronic obstructive pulmonary diseases (prevalence 6.5% [63]) with around EUR 16 billion of yearly costs (2008: EUR 13 billion [64,65]) or diabetes mellitus on the higher end (prevalence 9.9% [66]) with cost of roughly EUR 52 billion per year (2007: EUR 42 billion [67]).

As clear limitations to the study, one has to mention the focus on one study site. The study was performed in Berlin and therefor had only participants from the city and the regions around. This may limit the reliability of the data especially for the patients living in rural areas. Furthermore, we did not weight different scenarios within our cost estimations for their frequency of occurrence, as stated above. Therefore, the mean costs represent only a rough estimation of the true costs for the healthcare system. Finally, the estimations for the economic costs can be seen also only as a rough estimation, as not all professions in all social classes are represented. All these limitations should be addressed in future studies, i.e., they should be multicentric or even multinational and represent all socioeconomic classes.

## 5. Conclusions

With this study, we were able to show that tinnitus—in the view of socioeconomic costs—is not ‘just some’ symptom of multiple possible origin, but rather a severe condition that is comparable with other major diseases and it should be treated as such. More fine-grained future analyses might even conclude that—at least in the view of healthcare costs—tinnitus should be seen as a disease of its own. In any case, against the background of the socioeconomic costs estimated here, the relevance of tinnitus to all societies—not only in Germany—seems obvious.

## Figures and Tables

**Figure 1 ijerph-19-10455-f001:**
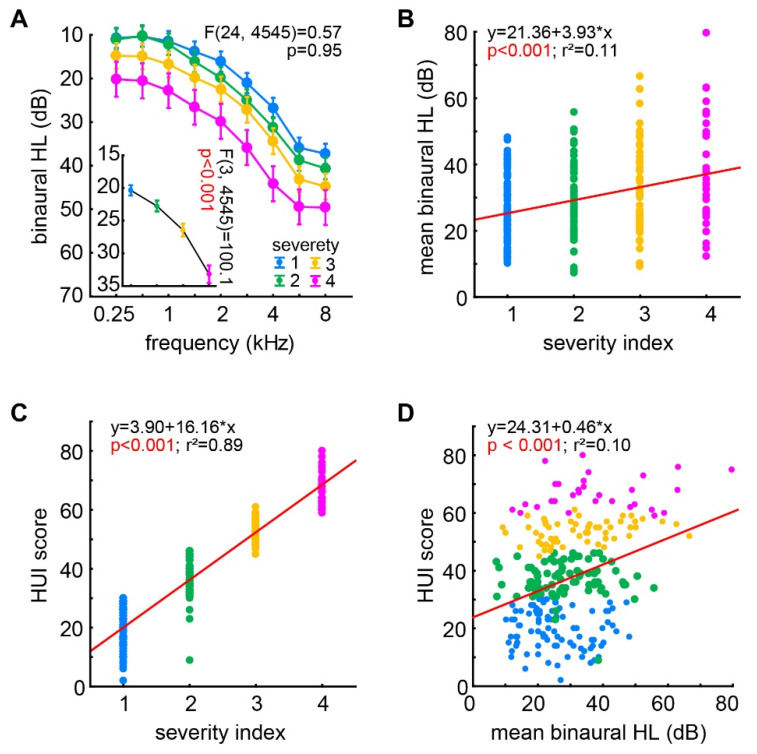
Relationship of HL, tinnitus severity index and HUI score. (**A**) Results of the two-factorial ANOVA of the patients’ binaural HL dependent on the factors ‘frequency’ and ‘tinnitus severity’. Given is the interaction plot of both factors with the F-statistics. The inset depicts the effect of the factor tinnitus severity (averaged over all frequencies). Note that all groups are significantly different from each other (Tukey post-hoc tests). (**B**) Significant multiple linear regression analysis of the individual mean binaural HL (averaged over all frequencies) and the tinnitus severity index. (**C**) Significant linear regression analysis of the individual HUI score and the tinnitus severity index. (**D**) Significant linear regression analysis of the individual HUI score and the mean binaural HL.

**Figure 2 ijerph-19-10455-f002:**
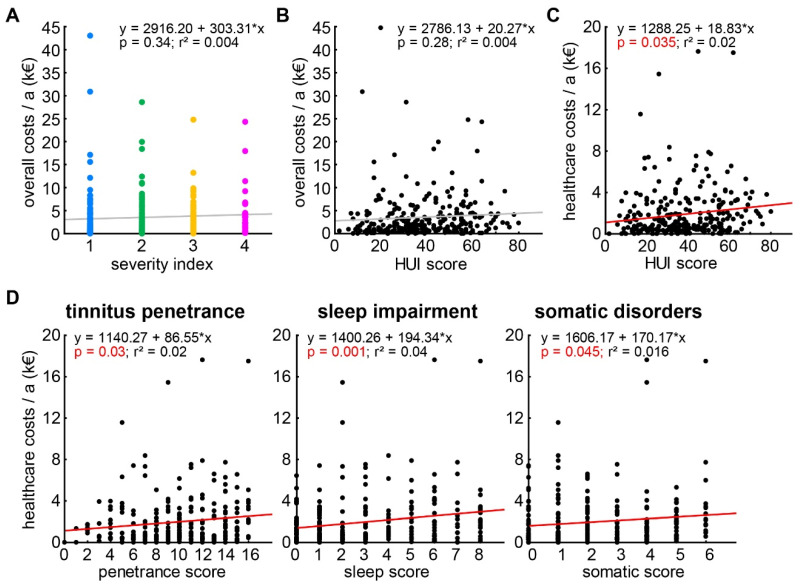
Multiple linear regression analyses of overall and healthcare costs with tinnitus severity index and HUI score/sub-scores. (**A**) No significant regression for the individual overall costs and the tinnitus severity index. Severity index color scheme as in Figure 1. (**B**) No significant regression (grey line) for the overall costs and the HUI score. (**C**) Significant linear regression (red line) for the healthcare costs and the HUI score. (**D**) Significant linear regressions (red lines) for the healthcare costs and the HUI sub-scores tinnitus penetrance, sleep impairment, and somatic disorders.

**Table 1 ijerph-19-10455-t001:** Overview of tinnitus severity of the 258 patients.

Severity Index	Number of Patients	Compensated/Uncompensated Tinnitus Patients
1	90	172
2	82	
3	56	86
4	30	

**Table 2 ijerph-19-10455-t002:** Costs of for the health care system for one visit.

Visit at Specific Medical Personal or Service	Mean Public Health Care Insurance Costs (EUR)	Mean Private Health Care Insurance Costs (EUR)
General practitioner (GP)	13.60	37.12
House call GP	45.11	60.14
Emergency service	21.10	49.06
ENT specialist	56.48	74.62
ENT clinic	98.90	138.05
Neurologist	81.25	212.06
Audiologist	255.00	500.00
Psychiatrist	25.44	67.04
Psychotherapist	86.38	107.69
Occupational therapist	65.72	118.30
Dentist	72.46	102.26
Physiotherapist	30.64	51.55
Social worker	8.59	8.59
Medical officer	6.75	6.75

**Table 3 ijerph-19-10455-t003:** Overview of mean costs for the health care system in one year.

	Severity 1		Severity 2		Severity 3 + 4		All Patients	
	Mean Visits	Mean Costs (EUR)	Mean Visits	Mean Costs (EUR)	Mean Visits	Mean Costs (EUR)	Mean Visits	Mean Costs (EUR)
GP	9.69	198.67	13.26	229.79	16.70	278.34	13.21	238.64
House call GP	0.04	2.00	0.10	4.40	0.08	4.20	0.12	6.27
Emerg. service	-	-	0.10	2.06	0.14	2.94	0.12	3.90
ENT specialist	3.87	233.71	4.63	273.24	6.33	364.86	4.96	292.33
ENT clinic	3.56	396.52	2.54	198.50	1.67	147.10	2.59	249.48
Neurologist	0.53	60.77	0.59	47.56	1.67	196.89	0.93	101.55
Audiologist	1.07	413.56	1.12	345.85	1.35	446.51	1.17	401.47
Psychiatrist/	2.00	188.86	3.51	352.24	4.05	367.38	3.15	299.13
Psychotherapist	0.89	37.40	1.46	85.93	0.56	18.07	0.96	46.20
Occupat. thera.	0.27	17.53	0.05	3.21	0.56	44.02	0.29	21.72
Dentist	0.31	27.84	1.07	82.12	1.12	80.89	0.82	62.53
Physiotherapist	5.07	206.36	6.78	251.61	11.07	387.81	7.58	280.14
Social worker	-	-	-	-	0.70	5.99	0.23	1.99
Medical officer	0.09	0.60	0.15	0.99	0.14	0.94	0.14	0.94
Others	0.3	28.88	0.92	159.12	0.81	340.49	0.75	173.48
Prescr. Drugs *	0.17	25.98	0.18	19.57	0.31	39.26	0.22	28.28
Complete costs	**27.68**	**1838.69**	**36.46**	**2056.20**	**47.27**	**2725.69**	**38.92**	**2206.95**

* for details on prescribed drugs refer to Table 4.

**Table 4 ijerph-19-10455-t004:** Costs of prescribed drugs.

Active Ingredient or Drug^®^	Pack Size (No. of Pills)	Lowest Found Price (EUR)
Amitriptylin	50	14.35
Carbamazepin	200	29.13
Diazepam	50	12.57
Ginkgo biloba	60	28.79
Lorazepam	50	14.30
Magnesium	50	9.95
Oxazepam	50	12.59
Zinc sulphate	20	3.49
Zolpidem	20	14.29
Amioxid-neurax^®^	50	12.70
Betavert^®^	50	17.99
Cinnarizin^®^	50	28.52
Laif 900^®^	60	23.26
Mirtazapin^®^	50	24.32
Prednisolon^®^	50	16.35
Seroxat^®^	50	47.95
Venlafaxin^®^	20	20.90
Zoloft^®^	50	32.25

**Table 5 ijerph-19-10455-t005:** Results of multiple linear regression analyses of severity index and HUI scores vs. costs.

First Variable	Second Variable	Linear Equation	r^2^ Value	*p* Value
individual overall costs	Severity index	y = 2916.2 + 303.3 × x	0.004	0.34
	HUI score (complete)	y = 2786.1 + 20.2 × x	0.004	0.28
	HUI: emotions	y = 3080.8 + 47.6 × x	0.003	0.40
	HUI: cognition	y = 3273.8 + 45.1 × x	0.001	0.57
	HUI: penetrance	y = 3254.2 + 31.4 × x	0.0005	0.71
	HUI: auditory	y = 3093.5 + 82.0 × x	0.004	0.33
	HUI: sleep	y = 2766.5 + 258.2 × x	0.02	0.04
	HUI: somatic	y = 3261.5 + 131.2 × x	0.002	0.46
individual healthcare costs	Severity index	y = 1389.0 + 291.3 × x	0.014	0.058
	HUI score (complete)	y = 1288.3 + 18.8 × x	0.02	0.035
	HUI: emotions	y = 1731.5 + 26.6 × x	0.004	0.33
	HUI: cognition	y = 1652.1 + 53.9 × x	0.008	0.16
	HUI: penetrance	y = 1140.3 + 86.6 × x	0.02	0.03
	HUI: auditory	y = 1690.9 + 54.0 × x	0.007	0.18
	HUI: sleep	y = 1400.3 + 194.3 ×x	0.04	0.001
	HUI: somatic	y = 1606.2 + 170.2 × x	0.016	0.045
individual private costs	Severity index	y = 396.1 + 62.1 × x	0.002	0.51
	HUI score (complete)	y = 259.2 + 7.1 × x	0.006	0.20
	HUI: emotions	y = 314.5 + 21.1 × x	0.006	0.21
	HUI: cognition	y = 472.9 + 8.8 × x	0.0005	0.71
	HUI: penetrance	y = 495.5 + 3.5 × x	0.0001	0.89
	HUI: auditory	y = 302.0 + 39.5 × x	0.01	0.11
	HUI: sleep	y = 342.7 + 60.5 × x	0.01	0.16
	HUI: somatic	y = 311.9 + 97.7 × x	0.01	0.073
individual economic costs	Severity index	y = 1131.1 − 50.1 × x	0.0002	0.83
	HUI score (complete)	y = 1238.7 − 5.6 × x	0.0007	0.68
	HUI: emotions	y = 1034.7 − 0.1 × x	<0.0001	0.99
	HUI: cognition	y = 1148.8 − 17.6 × x	0.0003	0.77
	HUI: penetrance	y = 1618.4 − 58.6 × x	0.003	0.35
	HUI: auditory	y = 1100.6 − 11.5 × x	0.0001	0.85
	HUI: sleep	y = 1023.5 + 3.4 × x	<0.0001	0.97
	HUI: somatic	y = 1343.4 − 132.7 × x	0.001	0.31

Note: Green numbers highlight *p* values that indicate a tendency for having a linear regression, red numbers highlight significant linear regressions.

**Table 6 ijerph-19-10455-t006:** Results of general linear model analyses of severity index and HUI scores vs. the different costs.

First Variable	Second Variable	F Value of Slope	R^2^ Value	*p* Value
individual overall costs	Severity index	0.44	0.005	0.72
	HUI score (complete)	1.19	0.005	0.27
	HUI: emotions	0.70	0.003	0.40
	HUI: cognition	0.32	0.001	0.57
	HUI: penetrance	0.14	0.0005	0.71
	HUI: auditory	0.97	0.004	0.33
	HUI: sleep	4.24	0.016	0.04
	HUI: somatic	0.55	0.002	0.46
individual healthcare costs	Severity index	1.66	0.02	0.18
	HUI score (complete)	4.48	0.017	0.035
	HUI: emotions	0.94	0.004	0.33
	HUI: cognition	1.95	0.008	0.16
	HUI: penetrance	4.54	0.018	0.034
	HUI: auditory	1.81	0.007	0.18
	HUI: sleep	10.57	0.040	0.001
	HUI: somatic	4.05	0.016	0.045
individual private costs	Severity index	1.59	0.018	0.19
	HUI score (complete)	1.65	0.006	0.20
	HUI: emotions	1.58	0.006	0.21
	HUI: cognition	0.14	0.0005	0.71
	HUI: penetrance	0.02	<0.0001	0.89
	HUI: auditory	2.57	0.01	0.11
	HUI: sleep	2.63	0.01	0.11
	HUI: somatic	3.24	0.01	0.07
individual economic costs	Severity index	0.34	0.004	0.79
	HUI score (complete)	0.17	0.0006	0.68
	HUI: emotions	0.00	0.00	1.00
	HUI: cognition	0.09	0.0003	0.77
	HUI: penetrance	0.88	0.003	0.35
	HUI: auditory	0.03	0.0001	0.85
	HUI: sleep	0.001	<0.0001	0.97
	HUI: somatic	1.04	0.004	0.31

Note: Green numbers highlight *p* values that indicate a tendency for having a dependency of both factors, red numbers highlight significant dependencies.

**Table 7 ijerph-19-10455-t007:** Results (median [interquartile range]) of Mann–Whitney U tests of yearly socioeconomic costs for compensated and uncompensated tinnitus patients.

	Compensated Tinnitus Patients	Uncompensated Tinnitus Patients	*p* Value
Overall costs	3657.40 [2742.72, 6089.08]	4912.02 [3314.84, 6346.34]	0.007
Healthcare costs	1857.64 [1430.65, 3270.50]	2835.04 [1668.20, 4272.80]	0.005
Private costs	153.54 [113.03, 439.46]	242.92 [135.25, 541.48]	0.009
Economic costs	2250.57 [0, 4142.90]	2158.62 [0, 5031.60]	0.34

Note: red *p* values highlight significant Mann–Whitney U tests.

## Data Availability

The data can be requested from the corresponding author.

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
