# Peer review of "Estimation of Tinnitus-Related Socioeconomic Costs in Germany"

_ijerph, 2022, doi:10.3390/ijerph191610455_

Round 1

Reviewer 1 Report

1. In the introduction part, Authors presented only one example of research on the socioeconomic costs of tinnitus and also without indication why this example (Dutch case) is so important (what was the argument to present this one) lines 36-42.

2. Some more information about Charité university hospital Berlin and 258 analyzed patients could be presented  in purpose to show how this sample is representative in the context of the total amount of total suffering from tinnitus (which is mentioned only in %  - line 25) and why this hospital was chosen. 

3. It is difficult to get know the idea of questionnare (questions) as they are in German (of course people in Germany must be asked in their mother language). 

4. How the reliability and accuracy of this questionnaire was measured and tested ? 

5. Elements of socioeconomics costs are not clearly specified. 

6. Results - cost of different scenarios could be presented in the form of table as they are not clearly presented. 

7. In the conclusion , the limitations and direction of future research should be mentioned. 

8. In the discussion part - more comparison should be made with the results of other research not only with the Dutch study results.

Author Response

  1. In the introduction part, Authors presented only one example of research on the socioeconomic costs of tinnitus and also without indication why this example (Dutch case) is so important (what was the argument to present this one) lines 36-42.

We added other studies investigating the socioeconomic costs of tinnitus to the introduction (p 3, l 18f) and added a sentence for the Dutch study.

  1. Some more information about Charité university hospital Berlin and 258 analyzed patients could be presented  in purpose to show how this sample is representative in the context of the total amount of total suffering from tinnitus (which is mentioned only in %  - line 25) and why this hospital was chosen. 

The Charité is one of the largest hospitals in Europe and also rated as the best in Europe (regarding to Newsweek). Independent of that, our colleagues work and started their investigation there. I don’t know, what else to say about the choice of the hospital. All relevant numbers of the patients are already given in the first section of the results (Patient statistics.

  1. It is difficult to get know the idea of questionnare (questions) as they are in German (of course people in Germany must be asked in their mother language). 

We apologize for this; we translated the questions for better intelligibility.

  1. How the reliability and accuracy of this questionnaire was measured and tested? 

We added the reliability measurement of Crombach’s Alpha to the statistics section of the Methods.

  1. Elements of socioeconomics costs are not clearly specified. 

We are not completely sure, what you refer to. We clearly state in the section Calculation of socioeconomic costs what specific factors are taken into account. These include public health care, private costs and economic loss (cf. p 3, l 25). We removed the “socio” from “socioeconomic” when explaining the economic part (e.g. p 5, l 16).

  1. Results - cost of different scenarios could be presented in the form of table as they are not clearly presented. 

We added Supplementary Table 1 to the manuscript for clarification.

  1. In the conclusion , the limitations and direction of future research should be mentioned. 

We added a limitation and future studies paragraph at the end of the discussion.

  1. In the discussion part - more comparison should be made with the results of other research not only with the Dutch study results.

We added the requested comparison in the paragraph starting on page 16, line 6.

Reviewer 2 Report

General remarks

The title of the manuscript perfectly identifies its purpose. The socioeconomic costs, to which I would add the mental health costs, associated with the occurrence of tinnitus, lead the authors to suggest that this symptom should be seen as a disease of its own. Interestingly, for someone like me who, actually, suffers from tinnitus -- which will not influence this review report of mine -- this is a conclusion I had already reached. In fact, as the World Health Organization includes Back and Neck Pain in the group M. Musculoskeletal diseases, perhaps it should be possible to include tinnitus in a group of diseases, eventually those of Ear, Nose and Throat (ENT).

Specific remarks

Literally, continuing what has just been said, I would start by suggesting that the manuscript be proof-read, in order to avoid some grammatical flaws, such as, for example:

·    Before using the acronym ENT, it must be defined (page 1: 25). Of course, authors, by their training, use it without this need, but not all readers will be familiar with what it means.

·        The last sentence of the first paragraph of the introduction, i.e., “Additionally, the phantom sound can lead to insomnia, psychological disorders or, for the most severe cases, even suicide attempts [11; 12; 13], all leading to the need of additional medical” appears to be incomplete.

·        In Figures 1 (page 5) and 2 (page 11) there is a typo in “severety”.

I also recommend extending the proof-reading to the list of references in order to format all of them as required by the Journal. To put is clearer, in reference [5] we have “J.A.M. Henry, M.; Gilbert, A.,”, whereas in reference [7] we have “J.J. Nelson, and K. Chen,”.

Continuing with the references, will the authors not want to read another – to which I have no kind of relationship -- that seems to me of undeniable interest, i.e. Trochidis, I.; Lugo, A.; Borroni, E.; Cederroth, C.R.; Cima, R.; Kikidis, D.; Langguth, B.; Schlee, W.; Gallus, S. Systematic Review on Healthcare and Societal Costs of Tinnitus. Int. J. Environ. Res. Public Health 2021, 18, 6881. https://doi.org/10.3390/ijerph18136881?

I also recommend that, in the concluding section -- which would change its name -- the authors acknowledge the limitations of their work, possibly as clues for future analysis.

I welcome the fact that the authors have made the questionnaire available as supplementary material. However, with all due respect for the German language, I suggest that it be translated into English.

I verify that, systematically, the authors use linear regression models, in which the explanatory variable is the severity index, which is, by construction, of a discrete nature. Plainly, the estimated regression models end up considering that the values of the explanatory variable respect a, let us say, (continuous) mathematical relationship between them. To be clearer, when the severity index goes, for instance, from 1 to 2, this means that the severity has doubled. Is this exactly what these 2 index values are meant to say? If not, the results of linear regression models should be treated very carefully.

Author Response

Specific remarks

Literally, continuing what has just been said, I would start by suggesting that the manuscript be proof-read, in order to avoid some grammatical flaws, such as, for example:

Before using the acronym ENT, it must be defined (page 1: 25). Of course, authors, by their training, use it without this need, but not all readers will be familiar with what it means.

The last sentence of the first paragraph of the introduction, i.e., “Additionally, the phantom sound can lead to insomnia, psychological disorders or, for the most severe cases, even suicide attempts [11; 12; 13], all leading to the need of additional medical” appears to be incomplete.

Thank you, we revised the manuscript accordingly.

In Figures 1 (page 5) and 2 (page 11) there is a typo in “severety”.

          We corrected it to severity.

I also recommend extending the proof-reading to the list of references in order to format all of them as required by the Journal. To put is clearer, in reference [5] we have “J.A.M. Henry, M.; Gilbert, A.,”, whereas in reference [7] we have “J.J. Nelson, and K. Chen,”.

Continuing with the references, will the authors not want to read another – to which I have no kind of relationship -- that seems to me of undeniable interest, i.e. Trochidis, I.; Lugo, A.; Borroni, E.; Cederroth, C.R.; Cima, R.; Kikidis, D.; Langguth, B.; Schlee, W.; Gallus, S. Systematic Review on Healthcare and Societal Costs of Tinnitus. Int. J. Environ. Res. Public Health 2021, 18, 6881. https://doi.org/10.3390/ijerph18136881?

The references have been corrected and expanded. The suggested paper is now also discussed.

I also recommend that, in the concluding section -- which would change its name -- the authors acknowledge the limitations of their work, possibly as clues for future analysis.

          We added a limitation section as the last paragraph of the discussion.

I welcome the fact that the authors have made the questionnaire available as supplementary material. However, with all due respect for the German language, I suggest that it be translated into English.

          We now provide the English translation.

I verify that, systematically, the authors use linear regression models, in which the explanatory variable is the severity index, which is, by construction, of a discrete nature. Plainly, the estimated regression models end up considering that the values of the explanatory variable respect a, let us say, (continuous) mathematical relationship between them. To be clearer, when the severity index goes, for instance, from 1 to 2, this means that the severity has doubled. Is this exactly what these 2 index values are meant to say? If not, the results of linear regression models should be treated very carefully.

The steps for the severity indices are defined by the scores of the questionnaires filled out by the patients. Index 1 has scores of <31 points, index 2 is ≤ 46 points, index 3 is ≤59 and index 4 is up to 84 points (maximum). The single steps are roughly equivalent to a “doubling” of the tinnitus burden, if one can claim something like this in a psychological questionnaire.

Reviewer 3 Report

Thank you for allowing me to review this article. I congratulate the authors on this work.  My main concern is the results of figure 2, page 11. 

The authors reported the correlation coefficient r square, which seems very close to zero.  I am convinced this is not a helpful result, and I suggest that the authors repeat this analysis using a Generalised regression model such as Poisson or Gamma regression. 

If they do so, It might be possible to see the dependence of the two variables under consideration. 

Author Response

The authors reported the correlation coefficient r square, which seems very close to zero.  I am convinced this is not a helpful result, and I suggest that the authors repeat this analysis using a Generalised regression model such as Poisson or Gamma regression.

If they do so, It might be possible to see the dependence of the two variables under consideration.

Thank you for your suggestion. We added the GLM statistics as additional analyses to the result section regarding Figure 2 and summarized them in the new Table 6.

Reviewer 4 Report

This is an interesting article in which the authors examine the Estimation of tinnitus related socioeconomic costs in Germany.  The comments below are made with the intent to strengthen the manuscript in order to improve its suitability for publication.  

Abstract:  This will need careful attention following changes to other aspects of the paper.  I do not see the sections clearly on the paper.  

Introduction:  On line 33 seems to be incomplete. 

Methods:  Please clarify why was analyzed the dependence of the binaural HL on frequency and tinnitus 115 severity by a two-factorial ANOVA.

On results, Tables 5 and 6, please include the discussion on the red/green colors. 

The discussion might mention what is the novelty of the study. 

The authors do not include limitations of the study. Please include them. 

Author Response

This is an interesting article in which the authors examine the Estimation of tinnitus related socioeconomic costs in Germany.  The comments below are made with the intent to strengthen the manuscript in order to improve its suitability for publication.  

Abstract:  This will need careful attention following changes to other aspects of the paper.  I do not see the sections clearly on the paper.  

We are not completely sure, what you mean with this comment, but we revised the whole manuscript text for intelligibility.

Introduction:  On line 33 seems to be incomplete.

This is corrected now, we revised the manuscript text.

Methods:  Please clarify why was analyzed the dependence of the binaural HL on frequency and tinnitus severity by a two-factorial ANOVA.

We clarified it, as this variable is normally distributed.

On results, Tables 5 and 6, please include the discussion on the red/green colors.

We added a remark to Table 5, 6 (new) and 7.

The discussion might mention what is the novelty of the study.

          This is already mentioned in the first sentence of the discussion.

The authors do not include limitations of the study. Please include them.

This is now includes as last paragraph of the discussion.

Round 2

Reviewer 1 Report

There are huge improvements. All comments were taken into account. 

Author Response

Thank you.

We hope, you'll find the manuscript now suitable for publication.

Reviewer 3 Report

I still believe that your results in Figure 2 are problematic. 

The score is not a continuous variable and violates the normality assumption of the linear regression process. I have suggested you model your data with a generalised linear regression model and that you would get more realistic results. 

Do your think that graph c of healthcare costs vs HUI score is modelled appropriately? Suppose that you have a score of 20. At this value, you have an average healthcare cost of around 1310.  This is not a representative value for cases with an HUI score = 20, and thus your linear regression is not significant 

it seems that you have not followed my suggestion. Please explain. Please consult a statistician. 

Author Response

We apologize, if we misunderstood your request. We now also performed the general linear model approach for the data in Figure 2a and included the results in Table 6. This is the results table of the general linear regression analyses, you requested in your first review round – all regressions are presented there. As you can see there, the results are basically identical, with the exception of the lack of a trend for severity index and healthcare costs. This is now stated on page 14, line 25f. We agree, that the regression shown in e.g., Fig 2c are more of the quality of a point cloud. But we disagree, that the HUI parameters are not suitable for regression analyses. An example (at least for the HUI change, which we do not have in our data) can be seen in Francis, Howard W., et al. "Impact of cochlear implants on the functional health status of older adults." The Laryngoscope 112.8 (2002): 1482-1488. Nevertheless, we already mention in the text that the r² values are extremely low and therefor have to be taken with great caution (page 14, line 17f).

Independent on the outcome of the regression, these results are somewhat secondary to our main statement in this report.